# Temperature Threshold Values of Bone Necrosis for Thermo-Explantation of Dental Implants—A Systematic Review on Preclinical In Vivo Research

**DOI:** 10.3390/ma13163461

**Published:** 2020-08-06

**Authors:** Kristian Kniha, Nicole Heussen, Eugenia Weber, Stephan Christian Möhlhenrich, Frank Hölzle, Ali Modabber

**Affiliations:** 1Department of Oral and Maxillofacial Surgery, University Hospital RWTH Aachen, Pauwelstraße 30, 52074 Aachen, Germany; fhoelzle@ukaachen.de (F.H.); amodabber@ukaachen.de (A.M.); 2Department of Medical Statistics, University Hospital of Aachen, Pauwelsstraße 19, 52074 Aachen, Germany; nheussen@ukaachen.de; 3Center of Biostatistic and Epidemiology, Medical School, Sigmund Freud University, Freudplatz 3, 1020 Vienna, Austria; 4Institute of Heat and Mass Transfer, RWTH Aachen University, Augustinerbach 6, 52062 Aachen, Germany; eugenia.weber@rwth-aachen.de; 5Department of Orthodontics, University of Witten/Herdecke, Alfred-Herrhausen Str. 45, 58455 Witten, Germany; Stephan.Moehlhenrich@uni-wh.de

**Keywords:** bone, necrosis, thermal, heat, cryoinsult

## Abstract

Purpose: Very high or low temperatures will lead to bone damage. The objective of this review was to analyze threshold values for thermal bone necrosis. Methods: Histological animal studies evaluating thermal effects on bone necrosis were selected via electronic and hand searches in English and German language journals until 1 November 2019. The outcome measures were temperature-exposure intervals and laser settings effecting bone damage. Furthermore, investigated parameters were the bone-to-implant contact ratios (BIC) and infrabony pockets around dental implants after thermal treatment. For quality assessment of studies, the CAMARADES study quality checklist was applied. Results: A total of 455 animals in 25 animal studies were included for data extraction after screening of 45 titles from 957 selected titles of the MEDLINE (PubMed), The Cochrane Library, Embase and Web of Science search. The threshold values for bone necrosis ranged between 47 °C and 55 °C for 1 min. A threshold value for cryoinsult and laser treatment has not yet been defined. However, temperatures in the vicinity of 3.5 °C produce a histologically proven effect on the bone and in the surrounding tissue. At 50 °C for 1 min, BIC values significantly decreased and infrabony pockets increased. Bone quality had an influence on the outcome, as cancellous bone suffered higher bone damage from thermal treatment compared to cortical bone. Conclusion: No clear threshold value for bone necrosis is available according to the current literature for warm and cold stimuli. More in-depth and clinical studies are required to provide further insights in assessing the potential of thermal necrosis for implant removal.

## 1. Introduction

The exposure of bone to high temperatures will lead to bone damage and necrosis [1]. Not only heat, but also targeted cooling affects the bone structure and vital cells in the surrounding tissue [2,3]. A single cold stimulus at −10 °C inevitably leads to surrounding skin and bone necrosis. [2,3]. This is also called “cryoinsult”.

Bone remodeling is a continuous process in which old bone tissue is broken down by osteoclasts and re-formed by osteoblasts at the same or a different site [4]. This physiological repair mechanism is also effective in the event of thermal damage to the bone. Surgical procedures often involve the risk of overheating [4,5]. Cutting, rotating, and vibrating instruments are used regularly in contact with the hard and soft tissue under cooling [6]. As soon as the cooling is no longer sufficient, temperatures rise rapidly [7]. When drilling into the bone in particular, overheating of the drill hole must always be avoided, as otherwise there is a risk of subsequent implant loss [8]. Moreover, there are also far-reaching consequences. 

This review was also conducted in order to define a threshold bone necrosis level according to the current literature for further thermo-explantation research. If a dental implant has to be removed due to inflammation, there have been reports of the explantation being carried out with a targeted implant overheating [9,10]. In order to minimize bone damage resulting from implant explantation using drills and milling devices, there were several publications that used ultra-high frequency surgical devices for thermo-explantation [9,10]. Another study involved a CO_2_ laser as a thermal device for implant removal [11]. These surgical procedures are currently neither suitable nor approved for clinical application. The risk of bone necrosis seemed high and uncontrollable as the implants were unevenly heated without considering threshold levels [12,13,14]. 

Heat may be caused by surgical procedures with insufficient cooling, such as implant drilling and bone cutting, or electrical, water, and laser devices [15,16]. Authors treated the cancellous and cortical bone with several devices, such as heated fluids, electric thermal probes, laser devices, and heated implants in order to gain knowledge of bone behavior and remodeling on thermal irritations [1,17,18,19]. Heat generation during bone drilling, especially in implantology, is a well-investigated field [20]. Nevertheless, published threshold levels present widely varying values. A critical reflection of cadaveric models revealed that no blood flow was present in these studies. Therefore, the definition of reliable threshold temperature values requires in vivo investigations. The blood flow in the bone can cause faster heat dissipation [21]. Poor thermal tissue conductivity results in local heat accumulation. If the heat is dissipated very slowly, an extended exposure time will produce heat damage [22]. As clinical studies do not carry out histopathological analyses for ethical reasons, conclusions were therefore drawn from animal experiments.

The scope of this study was to review the pertinent literature systematically regarding results of various in vivo animal investigations evaluating threshold values for thermal bone necrosis. Our aim is to provide insights into the temperature and exposure time that produce thermal bone damage in order to prevent the development of jaw necrosis. Furthermore, laser settings effecting bone damage were evaluated.

## 2. Materials and Methods

The protocol for this systematic review was registered on PROSPERO (CRD42019141867). This systematic review was reported according to the Preferred Reporting Items for Systematic review and Meta-Analysis Protocols (PRISMA-P [23]) statement, using the Population, Intervention, Comparison and Outcome (PICO) method [24]. 

### 2.1. Focused Question

The focused research question (PICO) of this review was to define the threshold values for thermal bone necrosis in animal studies. 

### 2.2. Search Strategy

MEDLINE (PubMed), the Cochrane Library, Embase and Web of Science database searches were performed to find articles published in the English language up to and including 1 November 2019 (Figure 1). For the MEDLINE search, the following terms and combinations were applied: (animal study) OR in vivo OR histopathology AND thermal osteonecrosis) OR thermo necrosis) OR thermal bone damage) OR heat induced osteonecrosis) OR cryoinsult induced osteonecrosis. 

With regard to The Cochrane Library search, the following combinations were used: histopathology in All Text AND thermal osteonecrosis in All Text OR thermo necrosis in All Text OR thermal bone damage in All Text OR heat induced osteonecrosis in All Text OR cryoinsult induced osteonecrosis in All Text.

With regard to the Embase search, the following combinations were used: (‘animal study’: ti,ab,kw OR ‘in vivo study’: ti,ab,kw OR histopathology:ti,ab,kw) AND ‘thermal osteonecrosis’: ti,ab,kw OR ‘thermo necrosis’: ti,ab,kw OR ‘thermal bone damage’: ti,ab,kw OR ‘heat induced osteonecrosis’: ti,ab,kw OR ‘cryoinsultinduced osteonecrosis’: ti,ab,kw.

For the Web of Science search, the following terms and combinations were applied: topic: (animal study) and topic: (in vivo) and topic: (histopathology) and topic: (thermal osteonecrosis) or topic: (thermo necrosis) or topic: (thermal bone damage) or topic: (heat induced osteonecrosis) or topic: (cryoinsult induced osteonecrosis).

In addition, the electronic search was complemented by a manual search of the reference lists of all included full texts. For the electronic MEDLINE search, a reference management software (Endnote X 8.2, Thomson Reuters) was used. The obtained publications from The Cochrane Library search were also imported into the reference management software and finally screened.

### 2.3. Inclusion Criteria

The inclusion criteria for the studies were as follows:(1)In vivo animal studies(2)Studies investigating thermal bone damage by histopathology(3)Studies at all levels of evidence, except case reports and expert opinion(4)Studies reporting on at least one of the outcome measures(5)Language: German or English

### 2.4. Exclusion Criteria

Studies from which data on selected outcome variables could not directly be retrieved or calculated were not considered. Systematic reviews, studies with missing thermal input, in vitro studies, randomized clinical trials and other clinical studies and cadaver studies were excluded.

### 2.5. Selection of Studies

After elimination of duplicates, 2 calibrated reviewers (KK, AM) independently reviewed titles, abstracts, and full texts in accordance with the inclusion criteria. All titles were included in the abstract screening. If the information in the abstract was not clear enough for selection purposes, the full text was reviewed.

### 2.6. Data Extraction

Data extraction was independently performed on all included studies using data extraction tables. If data for individual parameters of the systematic review was sufficient, a meta-analysis was performed. Any disagreement with regard to inclusion and exclusion was resolved by discussion between the reviewers. In case of missing or unclear data, or if the information was still not sufficient for evaluation, the study was excluded for the present review (Table 1).

### 2.7. Parameters Were Classified as Follows

(1)Temperature and exposure time leading to bone damage(2)Laser settings producing bone damage(3)Bone-to-implant contact ratio (BIC) around implants after thermal treatment(4)Infrabony implant pockets after thermal treatment

### 2.8. Risk of Bias in Individual Studies 

For quality assessment of studies, the CAMARADES study quality checklist was applied [25]. The bias evaluation included sample size calculation, animal exclusion or the blinded assessment of outcome, blinded induction of the model, statement of potential conflicts, random allocation, compliance with the animal welfare regulations, and whether the studies were published in peer-reviewed journals.

### 2.9. Statistical Analysis

BIC around implants after thermal treatment and infrabony implant pockets after thermal treatment were considered as outcomes to describe differences between test and control groups after thermal treatment. Effect sizes of continuous outcomes for each study were reported as mean differences, along with 95% confidence intervals (CI We planned to conduct a meta-analysis only if studies were comparable, i.e., if treatments, participants, and the underlying clinical question are similar enough for pooling. To evaluate the statistical heterogeneity between studies, the Q-test of homogeneity and I2 statistics as a percentage of the total variability across studies were used. The significance level of the Q-test was set to 0.10, and I2 values were categorized as 25%, 50%, and 75% for low, moderate, and high heterogeneity, respectively [26]. All analyses were performed with RevMan 5.3.5 (Cochrane C., London, UK). Given the clinical heterogeneity across trials, we have abstained from summarizing the study specific effects into one overall effect. Results were expressed by effect sizes for each study and corresponding forest plots. 

## 3. Results

### 3.1. Study Characteristics 

After application of the inclusion criteria, 25 animal studies were selected for review (Table 2, Table 3 and Table 4) [1,2,12,14,17,18,27,28,29,30,31,32,33,34,35,36,37,38,39,40,41,42,43,44,45]. 

A total of 455 animals were evaluated across multiple different species, including 224 rats, 117 rabbits, 70 emus, 41 sheep, and 10 dogs. 18 studies assessed temperature/time intervals and seven studies investigated the effect of laser application on bone necrosis. For temperature control, 11 studies used thermocouples and two infrared thermography. Bony regions of interest were 8 mandibles, 6 femurs, 5 tibiae, 4 calvarias, and one maxilla and iliac crest. Histopathological parameters evaluated 24 cortical and 16 cancellous sites. Due to the quality of reported data and the high clinical heterogeneity between studies, no meta-analysis was performed; nevertheless, the effect estimates of each study regarding BIC (Bone-to-Implant Contact) around implants and infrabony implant pockets after thermal treatment were presented (Figure 2 and Figure 3). 

### 3.2. Temperature and Exposure Time Leading to Bone Damage 

Cold and warm stimuli lead to bone damage and should therefore be separated. Three studies [2,29,31] reported cryoprobes in an emu model. The aim was to induce femoral head necrosis with temperatures ranging from −273 °C to −10 °C. The exposure time varied between 15 s and 9 min, producing bone necrosis in all cases. The lowest necrosis volume was reported at a temperature of −10 °C and an exposure time of 9 min. Goetz et al., 2008 concluded that temperatures below 3.5 °C to 1 °C produced histologically proven bone necrosis. 

The remaining part of 15 studies analyzed the effect of heat on bone structure at temperatures ranging from 33 °C to 190 °C. Heat stimuli of 44 °C and 1 min had not caused any damages on bone [12]. First tissue reactions, such as hyperemia, started at 47 °C for 1 min. Bone resorption and dead osteocytes have been reported at temperatures up to 50 °C [13]; however, no long-term thermal bone damage could be evaluated due to bone remodeling [14]. Furthermore, Lundskog, 1972 declared a temperature of 50 °C and an exposure time of 30 s as the threshold value of bone necrosis. Yoshida et al., 2009 had reported the results of the calvarial bone study, in which 48 °C and 1 min of heat stimuli had caused apoptosis of osteocyte and it had taken 5 weeks to regain bone formation. 

From those studies, the heat boundary stimuli of bone necrosis, which is no relation with species and sites, are likely to be 48 to 50 °C for 1 min, and that is almost compatible with Langskog’s report. Nevertheless, one study [39] stated that the threshold value must be 55 °C with an exposure time of 1 min. According to Arnoldi et al., 2012, a high temperature of 180–190 °C for only a very short time of a few seconds did not lead to bone necrosis.

### 3.3. Laser Settings Producing Bone Damage

Stubinger et al. (2011) assessed that, with a cooled Er:YAG laser with an energy output of up to 1000 mJ/pulse and 12 Hz, no thermal damage resulted. One study assessing an Er:YAG laser (cooled) with the settings of 2 W and 20 Hz led to a small bone necrosis layer directly after irrigation [43]. On the other hand, a CO_2_ laser (not cooled) at 20 W and 2 kHz produced an active resorption by osteoclasts; however, the histologic changes had less than 40 pm layer thickness after 6 weeks [38]. Pourzarandian et al., 2004 compared the Er:YAG and CO_2_ lasers. For the Er:YAG laser, the results after 14 days were a carbonized tissue that covered the treated surface, spots of mineralization, and new bone formation in a percentage of the treated area. Compared to that, there was no significant new bone formation after application of the CO_2_ laser.

### 3.4. BIC around Implants after Thermal Treatment

A temperature elevation up to 38.9 °C for 4 min and 1-month follow-up [28] showed similar BIC values around treated implants (mean 43.1% SD 2.80) and untreated implants (45% SD 1.30). Higher temperature values [17,18] of 60 °C for 1 min led to reduced BIC values around heated implants (mean 25.42% SD 1.49 in cancellous bone, mean 27.23% SD 12.44 in cortical bone) versus untreated implants (mean 38.05% SD 1.38 in cancellous bone, mean 31.94% SD 18.10 in cortical bone). However, only the implants inserted in the cancellous bone [17] presented significant differences. Similarly, at a lower temperature of 50 °C for 1 min, the cancellous bone led to a significantly lower BIC value, whereas around implants that were inserted in more cortical bone [18] no deviating BIC values were evaluated. 

### 3.5. Infrabony Implant Pockets after Thermal Treatment

Higher temperature values [17,18] not only led to loss of bone contact, but also the infrabony pockets next to the treated implant increased. Pocket values ranged around heated implants (60 °C for 1 min) at a mean 3.11 SD 0.33 in cancellous bone, mean 1.07 SD 0.44 in cortical bone and around untreated implants at a mean 1.21 SD 0.16 in cancellous bone, mean 0.56 SD 0.49 in cortical bone. The infrabony pockets were significantly larger compared to the control group in the cancellous area of the iliac crest of the sheep at 50 °C for 1 min (Figure 3) [17]. In contrast, the cortical area of the sheep mandible did not present any significant differences between the test and control groups for the same temperature/time interval. On the other hand, at 60 °C for 1 min, treated implants showed larger infrabony pockets for both treated groups, the cortical and cancellous groups (Figure 3). However, only the implants inserted in the cancellous bone [17] presented significant differences. The bone quality has had an influence on the outcome, as the iliac crest presented greater amount of bone pockets compared to the mandible.

### 3.6. Risk of Bias in Individual Studies

For quality assessment of studies, the CAMARADES study quality checklist was applied (Figure 4).

## 4. Discussion

The purpose of this review was to define threshold values for thermal bone necrosis in order to assess the potential of thermal necrosis for implant removal. Based on our findings no clear threshold value for bone necrosis is available according to the current literature. The researchers focused on several parameters, such as temperature and exposure time leading to bone damage, laser settings producing bone damage, BIC values around implants, and infrabony implant pockets after thermal treatment.

In recent years, advanced water-cooling systems have been quite effective in reducing heat storage during implant drilling. Therefore, there is currently only a limited number of recent clinical studies on the topic of heat-induced bone necrosis and implant loss available. Several authors have investigated the temperature thresholds leading to jaw necrosis with widely varying results [12,14,30,32]. The differences in the values may be attributed in part to the diversity of the experiments and the many different influencing variables such as blood flow, bone structure, and more [27]. In 1983, Eriksson and Albrektsson stated a temperature of 47 °C with an exposure time of 1 min as the threshold value for bone damage [12]. This value reflects the lower limit of possible damage and corresponds to the threshold values published by Lundskog (50 °C with an exposure time of 30 s) [32]. Berman et al., 1984 concluded that cortical bone is more resistant to heat than cancellous bone [1]. Jacobs et al., 1972 did not explicitly aim to determine a threshold value; however, they produced osteonecrosis at lower temperatures [46]. Other studies describe damage at temperatures from 43 to 68 °C [41,47].

A similar effect to that of heating can be achieved by targeted cooling. Temperatures in the range of 1 °C to 3.5 °C produce a histologically proven effect on the bone and in the surrounding tissue (max. 0.7 mm) [2,3]. An isotherm of 3.5 °C was published by Goetz et al., 2008 which best corresponded to the boundary of the osteonecrotic lesions; however, this was a cadaver study without in vivo results [3]. This review focused only on in vivo animal studies. A single cold stimulus from −10 °C to −20 °C inevitably leads to surrounding skin and bone necrosis. 

Furthermore, it is known that osteocytes can be damaged by exposure to temperatures above 45 °C for 15 s [44]. The degree of damage depends on the temperature and the exposure time. Several threshold values for bone necrosis have been announced by multiple studies. In 1972, Lundskog claimed 50 °C for 30 s [32], and in 1953, Rouiller stated 55 °C for 1 min for the exact threshold boarder [39]. Both investigations took place in rabbits. However, Lundskog studied the cortical and cancellous tibia bone and Rouiller preferred the cortical calvaria, metatarsi, and radii. It may be hypothesized that in the same individual, different regions lead to divergent threshold values. Therefore, it remains unclear if these values are transferable to the jawbone, especially the human jawbone. 

Furthermore, defined threshold values for cold stimulus and laser treatment leading to bone necrosis have not yet been published. Bone irritations of −10 °C or a laser setting of a cooled Er:YAG laser with 20 Hz and 2 W evidently led to bone necrosis. In particular, it remains difficult to compare different laser devices, as there is a variety of variable and adjustment possibilities, such as fiber thickness, device, wavelength, distance between objects, energy output, and continuous-wave or pulsating settings.

A critical reflection on this review showed that not all studies used thermocouples for exact temperature measurement. Thermography and thermocouples were described as temperature control systems. Especially with thermocouples their position is finally decisive. Internal systems are located directly at the point of measurement, whereas external remote systems cannot detect the individual temperature loss over the individual distance [48]. Additionally, the reduced thermal conductivity and the inhomogeneous properties of the bone must be considered. In contrast, the infrared thermographic camera only records the surface temperature, so that the temperature at the actual measuring point can only be determined to a limited extent. The different threshold values could either be explained by the different measurement techniques, or the individual bone characteristics during examination.

Furthermore, due to different methods (e.g., different thermal devices, different species and bone areas), the overall threshold values may be considered as an average value. Additionally, the statistical results of this systematic review should be interpreted considering the high degree of study heterogeneity. Due to the heterogeneity of the current literate and in order to avoid misleading results no meta-analysis was performed. 

The current literature could not give a clear threshold to clinicians concerning heat generation and osteonecrosis because studies presented widely varying results. It is difficult to apply these numerical results directly to humans. Each animal species has an individual bone metabolism that can differ from humans. Up to now it is completely unclear whether an implant can be loosened in this temperature range without triggering a starving necrosis. In order to avoid thermal bone damage in humans, the limit of 47 °C for 1 min should not be exceeded. However, in order to further explore thermo-explantation, these numerical values can only be used as a guideline and before applied to humans further preclinical studies should follow. Based on these findings more preclinical studies regarding the effect of temperature and time intervals on the development of a limited bone necrosis within the range between 47 °C and 55 °C for 1 min are necessary.

## 5. Conclusions

It can be concluded that no clear threshold value for bone necrosis is available according to the current literature. The values ranged between 47 °C and 55 °C for 1 min. A threshold value for cryoinsult and laser treatment has not yet been defined. Bone quality had an influence on the outcome, as cancellous bone suffered higher bone damage compared to cortical bone. It is suggested that more in-depth and clinical studies are required to provide further insights in assessing the potential of thermal necrosis for implant removal.

## Figures and Tables

**Figure 1 materials-13-03461-f001:**
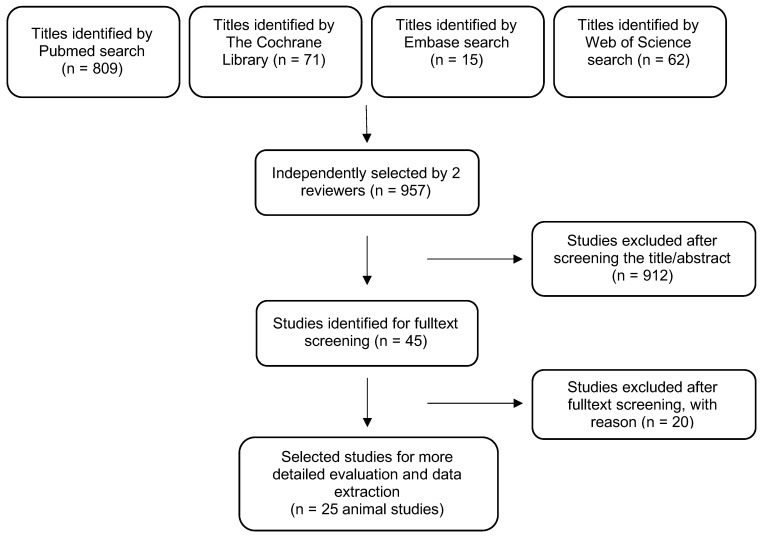
Search strategy overview.

**Figure 2 materials-13-03461-f002:**
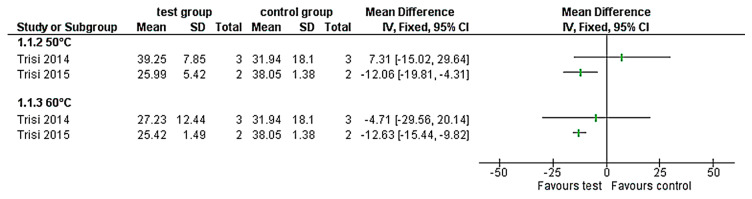
Forest-plots of the bone-to-implant contact ratio (%) values of the thermally treated implants (50 °C and 60 °C) compared with the control group.

**Figure 3 materials-13-03461-f003:**
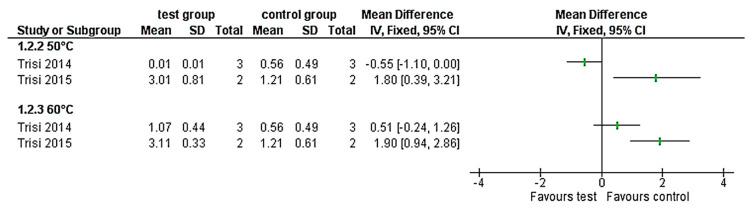
Forest-plots of the infrabony pockets (mm) around the implants that were thermally treated (50 °C and 60 °C) compared with the control group.

**Figure 4 materials-13-03461-f004:**
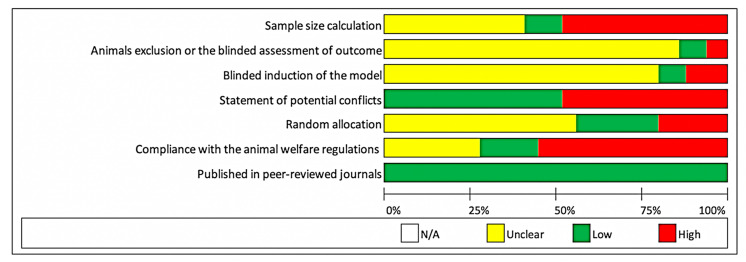
Quality assessment of studies using the CAMARADES study quality checklist.

**Table 1 materials-13-03461-t001:** Studies excluded after full-text screening.

Author (Year)	Reasons of Exclusion
Gholampour and Deh 2019	No histological evaluation
Fuchsberger 1988	No histological evaluation
Fontana et al., 2004	No histological evaluation
Connor and Hynynen 2004	No histological evaluation
Baker et al., 2011	No histological evaluation
Bonfield and Li 1968	No histological evaluation
Barnett 2001	No requested outcome measures
Myers et al., 1980	No requested outcome measures
Dolan et al., 2012	No requested outcome measures
Petersohn et al., 2008	No requested outcome measures
Posen et al., 1965	No requested outcome measures
Ivanenco et al., 2002	Only in vitro analysis
Lai et al., 2011	Only in vitro analysis
Van Egmod et al., 1994	Only in vitro analysis
Ryan et al., 1991	Only in vitro analysis
Danckwardt-Lillistrom 1969	Only in vitro analysis
Matthews et al., 1984	Only in vitro analysis
Franssen et al., 2008	No thermal values
Carvalho et al., 2011	No thermal values
Stubinger et al., 2009	No animal study

**Table 2 materials-13-03461-t002:** Included animal studies evaluating temperature parameters of –273 °C to 50 °C on bone damage (BIC = bone-to-implant contact ratio, n.s. = not specified, s = seconds, m = minutes, h = hours, d = days, w = weeks, and mon = months).

N	Author, Year	Species	Bone Area	N Animals	Thermal Application	Temperature Measurement System and Location	Temperature Max. in °C	Time	Effect on Bone	Follow up Time	BIC%
1	Fan et al., 2011	Emu	Femoral head cortical and cancellous	20	Liquid nitrogen and radiofrequency heating	Cryoprobe with external control system	−273	9 m nitrogen and 5 m heat	Necrosis	16 w	n.s.
2	Conzemius et al., 2002	Emu	Femoral head cortical and cancellous	22	Pressurized liquid nitrogen	n.s.	−196	15 s	Necrosis	6 w	n.s.
3	Goetz et al., 2011	Emu	Femoral head cortical and cancellous	28	Kirschner wire	Cryoprobe with two thermocouples at the exposed tip and one on the resistance-heated shaft	−50	9 m	1410 necrosis volume (mm^2^)	1 w	n.s.
							−40	9 m	1190 necrosis volume (mm^2^)	1 w	n.s.
							−30	9 m	1000 necrosis volume (mm^2^)	1 w	n.s.
							−20	9 m	700 necrosis volume (mm^2^)	1 w	n.s.
							−10	9 m	400 necrosis volume (mm^2^)	1 w	n.s.
4	Lye et al., 2011	Monkey	Mandible cortical and cancellous	6	Cemented endoprothesis	One external thermocouple at the test location	33.0	11 m	No thermal damage	3 m	n.s.
5	Yoshida et al., 2009	Rat	Calvaria cortical	120	Thermosimulator	Device with internal thermocouple	37	15 m	Control, TRAP-positive cells were reduced	1 w, 3 w and 5 w	n.s.
							43	15 m	TRAP-positive cells and ALP-positive cells were mostly absent on the bone surface after 1 w	1 w, 3 w and 5 w	n.s.
							45	15 m	Rising average of dead osteocyte layers	1 w, 3 w and 5 w	n.s.
							48	15 m	Apoptotic osteocytes were detected, high count dead osteocytes, no bone necrosis	1 w, 3 w and 5 w	n.s.
6	Calvo-Guirado et al., 2015	Dog	Mandible cortical and cancellous	6	Implant drill	Two external thermocouples next to the drill hole	38.9	up to 4 m	No thermal damage	1 m	mean 43.1% SD 2.80
										3 m	mean 64% SD 3.30
7	Mai et al., 2007	Sheep	Mandible cortical and cancellous	12	Frictional heat pins	n.s.	40	4 h	No thermal bone damage	2 w and 9 w	n.s.
8	Eriksson et al., 1984	Rabbit	Fibula cortical and cancellous	10	Heated saline solution	External thermocouple in the Thermostat	50	1 m	No thermal bone damage	n.s.	n.s.
9	Lundskog 1972	Rabbit	Tibia cortical and cancellous	n.s.	Electric thermal probe	Infrared thermography	50	30 s	Threshold necrosis	n.s.	n.s.

**Table 3 materials-13-03461-t003:** Included animal studies evaluating temperature parameters of 47 °C to 190 °C on bone damage (BIC = bone-to-implant contact ratio, n.s. = not specified, s = seconds, m = minutes, h = hours, d = days, w = weeks, and mon = months).

N	Author, Year	Species	Bone Area	N Animals	Thermal Application	Temperature Measurement System and Location	Temperature Max. in °C	Time	Effect on Bone	Follow up Time	BIC%	Infrabony Pockets Implant
10	Eriksson and Albrektsson 1983	Rabbit	Tibia cortical and cancellous	15	Heated implant chamber	Thermocouple inserted in the chamber with direct contact to the observed bone	47	1 m	Hyperemia, no vessel long term effects, 2 d slower fat cell resorption, slower new fat cell formation, slower bone resorption after 30 d and remodeling	up to 4 w	n.s.	n.s.
							47	5 m	Hyperemia, 5 d vessel diameter increase, 2 d fat cell resorption, 3 w new fat cell formation and bone resorption, 30 d up to 30% bone resorption and remodeling	up to 4 w	n.s.	n.s.
							50	1 m	Hyperemia, 41° blood flow increase, 50° blood flow stop, 2 d fat cell resorption, 10 d revascularization, 3 w new fat cell formation and bone resorption	up to 4 w	n.s.	n.s.
							55–70	30 s	Necrosis	n.s.	n.s.	n.s.
11	Eriksson et al., 1982	Rabbit	Tibia cortical and cancellous	5	Heated implant chamber	Thermocouple inserted in the chamber with direct contact to the observed bone	53	1 m	Stop of blood flow up to 3–4 w revascularization, after 2 d connective tissue injury, 6–8 w formation of new fat cells, after 5 w bone remodeling and osteogenesis	up to 10 w	n.s.	n.s.
12	Thompson 1958	Dog	Mandible cortical and cancellous	n.s.	Frictional heat pins	Infrared thermography	40–67	n.s.	Osteocyte degeneration and hyperemia	n.s.	n.s.	n.s.
13	Berman et al., 1984	Rabbit	Tibia cortical	18	Heated isotonic fluid	Internal thermocouple in the bath	45–55	1 m	Inflammation and fibrous tissue scar	1, 2 and 3 w	n.s.	n.s.
							70	1 m	Necrosis	1, 2 and 3 w	n.s.	n.s.
14	Rouiller 1953	Rabbit	Calvaria, metatarsi and Radii cortical	27	Heated metal	n.s.	46	5 m	Proliferation	24 h	n.s.	n.s.
							55	1 m	Threshold necrosis	24 h	n.s.	n.s.
15	Tillotson et al., 1989	Dog	Femur cortical and cancellous	4	Radiofrequency electrodes	Thermocouple at the tip of the probe	80	up to 4 m	Bone necrosis diameter 0.9. 1.3 cm, muscle necrosis	6 w	n.s.	n.s.
16	Trisi et al., 2015	Sheep	Iliac crest cancellous	2	Heated electronic device	Internal thermocouple in the device	50	1 m	Peri-implant bone loss, low density bone is more subject to heat-induced injury	2 m	n.s.	n.s.
							60	1 m	Peri-implant bone loss, influence on the osseointegration	2 m	n.s.	n.s.
17	Trisi et al., 2014	Sheep	Mandible cortical	3	Heated electronic device	Internal thermocouple in the device	50	1 m	No bone resorption, no threshold to heat-induced injury	2 m	mean 25.99% SD 5.42	mean 3.01 SD 0.81
							60	1 m	Peri-implant bone loss	2 m	mean 25.42% SD 1.49	mean 3.11 SD 0.33
18	Arnoldi et al., 2012	Rabbit	Femur cortical and cancellous	10	Ultrasonic energy pins	None	180–190	several seconds	—5 d cortical sites were more sensitive compared to cancellous sites, —4 w. No signs of tissue degeneration, new bone formation	5 d and 4 w	mean 39.25% SD 7.85	mean 0.01 SD 0.01
											mean 27.23% SD 12.44	mean 1.07 SD 0.44

**Table 4 materials-13-03461-t004:** Included animal studies evaluating laser parameters on bone damage (n.s. = not specified, s = seconds, m = minutes, h = hours, d = days, w = weeks, and mon = months).

N	Author, Year	Species	Bone Area	N of Animals	Thermal Application	Temperature Measurement System	Laser Settings	Exposure Time	Effect on Bone	Follow up
19	Martins et al., 2011	Rat	Mandible cortical	20	Er:YAG laser (cooled)	n.s.	300 mJ/6 Hz, 350 mJ/6 Hz, and 400 mJ/6 Hz	10 s	Similar healing pattern, 7 d thermal damage with thin layer of surface carbonization, 60 d amorphous layer persisted, 90 d no residual thermal damage was observed	up to 90 d
20	Nakamura et al., 1999	Rat	Mandible cortical	30	Excimer laser	n.s.	1.0 J/pulse and 10 Hz, 0.12 W, wavelength 193 nm (pulse duration 10–12 nsec, energy density 270 J/cm^2^ for 90 s with 34 °C and 360 J/cm^2^ for 120 s with 45 °C)	90 s and 120 s	Neither carbonization nor necrotic zone was observed at the surrounding tissue. Some vacuolar degeneration of osteocyte adjacent to the defect was observed. Minimal thermal damage.	directly after irrigation
21	Pourzarandian et al., 2004	Rat	Calvaria cortical	24	Er:YAG laser (cooled)	n.s.	100 mJ/pulse and 10 Hz, 1 W, wavelength 2.94 μm (pulse duration of 200 μs)	1.6 mm/s	—10 min a granular precipitate, red blood cells in aggregates of varying density predominant, thin fibrillar strands and inflammatory cells between cell aggregates.	up to 14 d
									—6 h many polymorphonuclear leukocytes, —1 d decrease in the number of red blood cells. Polymorphonuclear leukocytes and macrophages.	
									—3 d maturation of the fibrin clot, and a reduced red blood cell population. The polymorphonuclear leukocyte population increased and fibroblasts. Phagocytosis and angiogenesis.	
									—7 d cell-rich granulation tissue contained fibroblasts, and clusters of osteoblasts closely adapted to the bone, spots of mineralization identified, collagen fibrils surrounding osteoblasts.	
									—14 d new bone formation.	
					CO_2_ laser (not cooled)	n.s.	4 W, continuous wavelength of 10.6 μm	2.5 mm/sec	—10 m a carbonized layer with microcracks and porosities, a zone of thermal necrosis.	up to 14 d
									—6 h many polymorphonuclear leukocytes.	
									—1 d the population of polymorphonuclear leukocytes increased.	
									—3 d polymorphonuclear leukocytes predominant, proceeding to clear the necrotic or carbonized material. Healing started	
									—7 d fibrillar strands organized.	
									—14 d carbonized tissue still covered the treated surface. spots of mineralization, percentage of the area of new	
									—bone formation of Er:YAG laser and no significant new bone formation of CO_2_ laser.	
22	Rayan et al., 1992	Rabbit	Femur cortical and cancellous	20	CO_2_ laser (not cooled)	n.s.	20 W, 2 kHz spike pulse for 10 s. The pulse duration is 0.1 ms	10 s	4- and 6-week bony healing. Superficial zone on the inner cortex carbonization, residuals from vaporized tissue. bone resorption. encapsulated by reactive cells, evidence of new bone formation, deeper region of cellular thermal damage and bone necrosis but without vaporization. Active resorption by osteoclasts, histologic changes had less than 40 pm	4 w and 6 w
23	Stubinger et al., 2011	Sheep	Tibia cortical an cancellous	24	Er:YAG laser (cooled)	n.s.	1000 mJ/pulse and 12 Hz, energy density 157 J/cm^2^ (pulse duration of 300 μs, applied water spray level was 40–50 mL/m)	2 mon 245.33 s SD 29.9.3 mon 211.17 s SD 45.1	No thermal damage	2 mon and 3 mon
24	Wang et al., 2005	Rabbit	Mandible and maxilla cortical	12	Er:YAG laser (cooled)	n.s.	20 Hz, 2 W (emitting at 2.78 lm, pulsed with a duration of 140–200 ls, tip used was 400 lm in diameter and 8.0 mm in length)	10 s	70–90 micron bone necrosis (mean 30)	directly after irrigation
25	Yoshida et al., 2009	Rat	Calvaria cortical	30	Contact Er:YAG laser (not cooled)	n.s.	115 mJ/pulse and 10 Hz. Contact focused irradiation (energy density: 40.7 J/cm^2^/pulse)	1 cm per 3 s	No major thermal changes were noted around the ablation defect	up to 6 m
					Non-contact Er:YAG laser (not cooled)	n.s.	115 mJ/pulse and 10 Hz. Non-contact defocused irradiation (6.6 J/cm^2^/pulse)	1 cm per 3 s	No major thermal changes were noted around the ablation defect	up to 6 m

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
