# Peer review of "Temperature Threshold Values of Bone Necrosis for Thermo-Explantation of Dental Implants—A Systematic Review on Preclinical In Vivo Research"

_materials, 2020, doi:10.3390/ma13163461_

Round 1
Reviewer 1 Report
Thank you very much for presenting this review of the current literature. Please correct the bibliography as it does not follow the Materials format. Improve the resolution of the graphs of the manuscript and the flowchart. The figures are pixelated. They look blurry. Improve the writing of the search strategy for articles in Material and methods. It is not clear. Articles were not searched in other databases such as Embase or WOS ?? It is a very interesting article and very well written.
Author Response
Dear Editors, dear reviewers,
Thank you very much for the opportunity of the major revision and the helpful comments on the manuscript “Temperature threshold values of bone necrosis for thermo-explantation of dental implants – a systematic review on preclinical in vivo research " ID: materials-878575. Below you will find a checklist regarding the comments and your desired changes point by point. All notes and comments were worked in new version and were highlighted with “track change”.
Reviewer 1:
Concerns of reviewer 1:
Thank you very much for presenting this review of the current literature. Please correct the bibliography as it does not follow the Materials format. Improve the resolution of the graphs of the manuscript and the flowchart. The figures are pixelated. They look blurry. Improve the writing of the search strategy for articles in Material and methods. It is not clear. Articles were not searched in other databases such as Embase or WOS ?? It is a very interesting article and very well written.
Response to reviewer: Thank you for your helpful comments. The bibliography was corrected. Furthermore, the resolutions of the figures of the manuscript and the flowchart were improved and corrected. In order to provide an overview of the literature as complete as possible, we have checked and included the Embase and Web of Science databases.
Text change: Bibliography, Figures 1,2 and 3, Embase and Web of Science databases were checked and included.
Thank you for your support,
Kind regards,
Kristian Kniha, DDS, Priv. Doz.
Reviewer 2 Report
- In this study, the results of experiments on different bones in different species of animals were compared. It is difficult to apply these numerical results directly to human beings, so it would be nice to discuss more about how to interpret these numerical results when applying them to humans.
- (Page 1, line 31-32) BIC values significantly increased? Not decreased?
- (Page 7, Tabel 3) Is '-47' correct? Please check again.
Author Response
Dear Editors, dear reviewers,
Thank you very much for the opportunity of the major revision and the helpful comments on the manuscript “Temperature threshold values of bone necrosis for thermo-explantation of dental implants – a systematic review on preclinical in vivo research " ID: materials-878575. Below you will find a checklist regarding the comments and your desired changes point by point. All notes and comments were worked in new version and were highlighted with “track change”.
Reviewer 2:
In this study, the results of experiments on different bones in different species of animals were compared. It is difficult to apply these numerical results directly to human beings, so it would be nice to discuss more about how to interpret these numerical results when applying them to humans.
(Page 1, line 31-32) BIC values significantly increased? Not decreased?
(Page 7, Tabel 3) Is '-47' correct? Please check again.
Response to reviewer: We improved the discussion, especially how to interpret these results when and if applying them to humans.
Additionally, on page 1, line 31-32 the BIC values significantly decreased, you are completely right, thank you.
On Page 7, Tabel 3, we corrected the caption to 47°.
Text change: Abstract: “At 50 °C for 1 min, BIC values significantly decreased and infrabony pockets increased.”
Discussion” It is difficult to apply these numerical results directly to humans. Each animal species has an individual bone metabolism that can differ from humans. Up to now it is completely unclear whether an implant can be loosened in this temperature range without triggering a starving necrosis. In order to avoid thermal bone damage in humans, the limit of 47°C for 1 minute should not be exceeded. However, in order to further explore thermo-explantation, these numerical values can only be used as a guideline and before applied to humans further preclinical studies should follow. Based on these findings more preclinical studies regarding the effect of temperature- and time intervals on the development of a limited bone necrosis within the range between 47 °C and 55 °C for 1 min are necessary.”
Thank you for your support,
Kind regards,
Kristian Kniha, DDS, Priv. Doz.
Reviewer 3 Report
Dear Authors,
The main assumption of the study is to find experimental data on the effect of temperature on the bone in vivo.
The article is interesting and worth publishing as it contains a compilation of important data of threshold values for thermal bone necrosis.
1.
I am not sure if the authors narrowed down their search using the password: "heat-induced"
OR " cryoinsult-induced"
In the case of : "thermal-induced" word "thermal" is well considered as separate keyword.
You also excluded human studies, but such data may be available for bone heating and water cooling.
2.
Line 116-118
Definition of the temperature measurement technique in the articles is, in my opinion, relatively too general.
There is no precise and critical description of the measurement technique used in the cited works. This forces the Reader to consult the source. Good review work provides ready-made data with a satisfactory level of accuracy.
It is worth presenting how others measured it and subjected to criticism and comparison. For example, how a thermocouple is used.
You say : "... not all studies used thermocouples for exact temperature measurement." S
so what do they use? thermography ? Is the application of a thermocouples sufficient? Its location is, after all, crucial.
Now we find out that the threshold does not exist or is very divergent. Meanwhile, we do not find out to what extent it may result from the measurement technique and to what extent from the properties of the examined bone, which, after all, cannot differ from each other beyond the values resulting from its elasticity and hardness properties (if drilling is used) and heat conduction.
3.
The study also lacks data on the problem of bone necrosis and implant loss in clinical practice. Thus, research motivation is not supported by clinical data, especially in recent years, when advances in water cooling are quite effective in reducing this problem.
Author Response
Dear Editors, dear reviewers,
Thank you very much for the opportunity of the major revision and the helpful comments on the manuscript “Temperature threshold values of bone necrosis for thermo-explantation of dental implants – a systematic review on preclinical in vivo research " ID: materials-878575. Below you will find a checklist regarding the comments and your desired changes point by point. All notes and comments were worked in new version and were highlighted with “track change”.
Reviewer 4:
Dear Authors,
The main assumption of the study is to find experimental data on the effect of temperature on the bone in vivo.
The article is interesting and worth publishing as it contains a compilation of important data of threshold values for thermal bone necrosis.
Response to reviewer: Thank you for this comment.
1.
I am not sure if the authors narrowed down their search using the password: "heat-induced"
OR " cryoinsult-induced"
In the case of : "thermal-induced" word "thermal" is well considered as separate keyword.
You also excluded human studies, but such data may be available for bone heating and water cooling.
Response to reviewer: Instead of “heat-induced" OR "cryoinsult-induced" we checked the databases again with separate keywords “heat induced" OR "cryoinsult induced", in order to provide an overview of the literature as complete as possible.
Text change: Methods and figure 1
2.
Line 116-118
Definition of the temperature measurement technique in the articles is, in my opinion, relatively too general.
There is no precise and critical description of the measurement technique used in the cited works. This forces the Reader to consult the source. Good review work provides ready-made data with a satisfactory level of accuracy.
It is worth presenting how others measured it and subjected to criticism and comparison. For example, how a thermocouple is used.
You say : "... not all studies used thermocouples for exact temperature measurement." S
so what do they use? thermography ? Is the application of a thermocouples sufficient? Its location is, after all, crucial.
Response to reviewer: Thank you we definitely agree, therefore next to the temperature measurement system column in tables 2 and 3 we added more information of the system and the measurement location. Additionally, we discussed if the application of thermography or thermocouples are sufficient.
Text change: Tables 2 and 3,
Discussion:” Thermography and thermocouples were described as temperature control systems. Especially with thermocouples their position is finally decisive. Internal systems are located directly at the point of measurement, whereas external remote systems cannot detect the individual temperature loss over the individual distance [48]. Additionally, the reduced thermal conductivity and the inhomogeneous properties of the bone must be considered. In contrast, the infrared thermographic camera only records the surface temperature, so that the temperature at the actual measuring point can only be determined to a limited extent.”
3.
Now we find out that the threshold does not exist or is very divergent. Meanwhile, we do not find out to what extent it may result from the measurement technique and to what extent from the properties of the examined bone, which, after all, cannot differ from each other beyond the values resulting from its elasticity and hardness properties (if drilling is used) and heat conduction.
Response to reviewer: Thank you for this comment. We added this explanation to the discussion.
Text change: “The different threshold values could either be explained by the different measurement techniques, or the individual bone characteristics during examination.”
4.
The study also lacks data on the problem of bone necrosis and implant loss in clinical practice. Thus, research motivation is not supported by clinical data, especially in recent years, when advances in water cooling are quite effective in reducing this problem.
Response to reviewer: We discussed recent in vivo data about this topic.
Text change: “In recent years, advanced water-cooling systems have been quite effective in reducing heat storage during implant drilling. Therefore, there is currently only a limited number of recent clinical studies on the topic of heat-induced bone necrosis and implant loss available.”
Thank you for your support,
Kind regards,
Kristian Kniha, DDS, Priv. Doz.
Round 2
Reviewer 2 Report
Thank you for revising your manuscript.
Nice work.
Reviewer 3 Report
Ok, the flaws have been fixed.